# Bridging Nonlinearities and Stochastic Regularizers with Gaussian Error Linear Units

**Dan Hendrycks**[*]
University of Chicago
dan@ttic.edu

**Kevin Gimpel**
Toyota Technological Institute at Chicago
kgimpel@ttic.edu

## Abstract

We propose the Gaussian Error Linear Unit (GELU), a high-performing neural network activation function. The GELU nonlinearity is the expected transformation of a stochastic regularizer which randomly applies the identity or zero map to a neuron's input. This stochastic regularizer is comparable to nonlinearities aided by dropout, but it removes the need for a traditional nonlinearity. The connection between the GELU and the stochastic regularizer suggests a new probabilistic understanding of nonlinearities. We perform an empirical evaluation of the GELU nonlinearity against the ReLU and ELU activations and find performance improvements across all tasks.

## 1 Introduction

Early artificial neurons utilized binary threshold units (Hopfield, 1982; McCulloch & Pitts, 1943). These hard binary decisions are smoothed with sigmoid activations, enabling a neuron to have a "firing rate" interpretation and to train with backpropagation. But as networks became deeper, training with sigmoid activations proved less effective than the non-smooth, less-probabilistic ReLU (Nair & Hinton, 2010) which makes hard gating decisions based upon an input's sign. Despite having less of a statistical motivation, the ReLU remains a competitive engineering solution which often enables faster and better convergence than sigmoids. Building on the successes of ReLUs, a recent modification called ELUs (Clevert et al., 2016) allows a ReLU-like nonlinearity to output negative values which sometimes increases training speed. In all, the activation choice has remained a necessary architecture decision for neural networks lest the network be a deep linear classifier.

Deep nonlinear classifiers can fit their data so well that network designers are often faced with the choice of including stochastic regularizer like adding noise to hidden layers or applying dropout (Srivastava et al., 2014), and this choice remains separate from the activation function. Some stochastic regularizers can make the network behave like an ensemble of networks, a pseudoensemble (Bachman et al., 2014), and can lead to marked accuracy increases. For example, the stochastic regularizer dropout creates a pseudoensemble by randomly altering some activation decisions through zero multiplication. Nonlinearities and dropout thus determine a neuron's output together, yet the two innovations have remained distinct. More, neither subsumed the other because popular stochastic regularizers act irrespectively of the input and nonlinearities are aided by such regularizers.

In this work, we bridge the gap between stochastic regularizers and nonlinearities. To do this, we consider an adaptive stochastic regularizer that allows for a more probabilistic view of a neuron's output. With this stochastic regularizer we can train networks without any nonlinearity while matching the performance of activations combined with dropout. This is unlike other stochastic regularizers without any nonlinearity as they merely yield a regularized linear classifier. We also take the expected transformation of this stochastic regularizer to obtain a novel nonlinearity which matches or exceeds models with ReLUs or ELUs across tasks from computer vision, natural language processing, and automatic speech recognition.

---

[*]Work done while the author was at TTIC. Code available at github.com/hendrycks/GELUs

## 2   GELUS AND THE STOCHASTIC 0-$I$ MAP

We create our stochastic regularizer and nonlinearity by combining intuitions from dropout, zoneout, and ReLUs. First note that a ReLU and dropout both yield a neuron's output with the ReLU deterministically multiplying the input by zero or one and dropout stochastically multiplying by zero. Also, a new RNN regularizer called zoneout stochastically multiplies inputs by one (Krueger et al., 2016). We merge this functionality by multiplying the input by zero or one, but the values of this zero-one mask are stochastically determined while also dependent upon the input. Specifically, we multiply the neuron input $x$ by $m \sim$ Bernoulli$(\Phi(x))$, where $\Phi(x) = P(X \leq x), X \sim \mathcal{N}(0,1)$ is the cumulative distribution function of the standard normal distribution. The distribution Bernoulli$(\Phi(x))$ appears in Gaussian Processes for classification (Houlsby et al., 2011) and the neuron's output is $xm$ giving $x$ or 0. Thus inputs have a higher probability of being "dropped" as $x$ decreases, so the transformation applied to $x$ is stochastic yet depends upon the input. Masking inputs in this fashion retains nondeterminism but maintains dependency upon the input value. A stochastically chosen mask amounts to a stochastic zero or identity transformation of the input, leading us to call the regularizer the SOI map. The SOI Map is much like Adaptive Dropout (Ba & Frey, 2013), but we refer to the regularizer as the SOI Map because adaptive dropout is used in tandem with nonlinearities. In section 4, we show that simply masking linear transformations with the SOI map exceeds the power of linear classifiers and competes with nonlinearities aided by dropout, showing that nonlinearities can be replaced with stochastic regularizers.

The SOI map can be made deterministic should we desire a deterministic decision from a neural network, and this gives rise to our new nonlinearity. The nonlinearity is the expected transformation of the SOI map on an input $x$, which is $\Phi(x) \times Ix + (1 - \Phi(x)) \times 0x = x\Phi(x)$. Loosely, this expression states that we scale $x$ by how much greater it is than other inputs. We now make an obvious extension. Since the cumulative distribution function of a Gaussian is computed with the error function, we define the Gaussian Error Linear Unit (GELU) as

$$\text{GELU}(x) = xP(X \leq x),$$

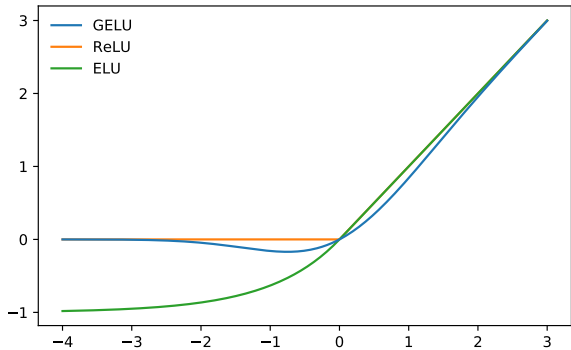

Figure 1: The GELU ($\mu = 0, \sigma = 1$), ReLU, and ELU ($\alpha = 1$).

where $X \sim \mathcal{N}(\mu, \sigma^2)$. Both $\mu$ and $\sigma$ are possibly parameters to optimize, but throughout this work we simply let $\mu = 0$ and $\sigma = 1$. Consequently, we do not introduce any new hyperparameters in the following experiments. In the next section, we show that the GELU exceeds the performance of ReLUs and ELUs across numerous tasks.

## 3   GELU EXPERIMENTS

We evaluate the GELU, ELU, and ReLU on MNIST classification (grayscale images with 10 classes, 60k training examples and 10k test examples), MNIST autoencoding, Tweet part-of-speech tagging (1000 training, 327 validation, and 500 testing tweets), TIMIT frame recognition (3696 training, 1152 validation, and 192 test audio sentences), and CIFAR-10/100 classification (color images with 10/100 classes, 50k training and 10k test examples). We do not evaluate nonlinearities like the LReLU because of its similarity to ReLUs (see Maas et al. (2013) for a description of LReLUs).

### 3.1   MNIST CLASSIFICATION

Let us verify that this nonlinearity competes with previous activation functions by replicating an experiment from Clevert et al. (2016). To this end, we train a fully connected neural network with GELUs ($\mu = 0, \sigma = 1$), ReLUs, and ELUs ($\alpha = 1$). Each 8-layer, 128 neuron wide neural

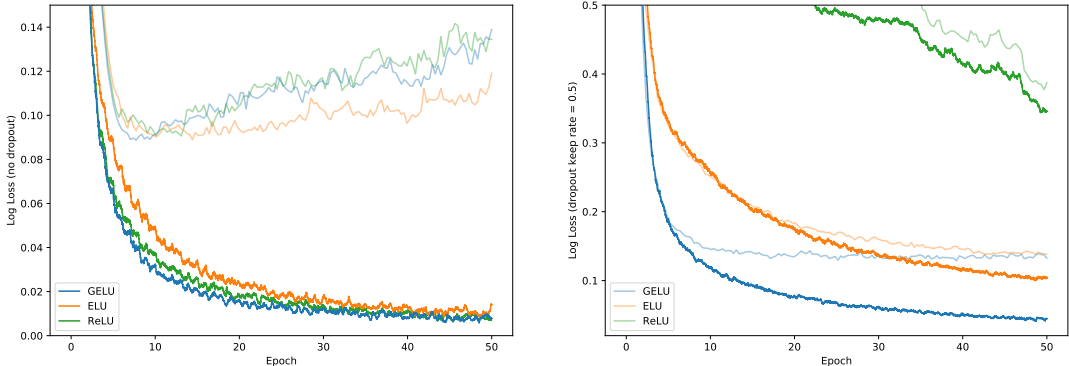

Figure 2: MNIST Classification Results. Left are the loss curves without dropout, and right are curves with a dropout rate of 0.5. Each curve is the the median of five runs. Training set log losses are the darker, lower curves, and the fainter, upper curves are the validation set log loss curves.

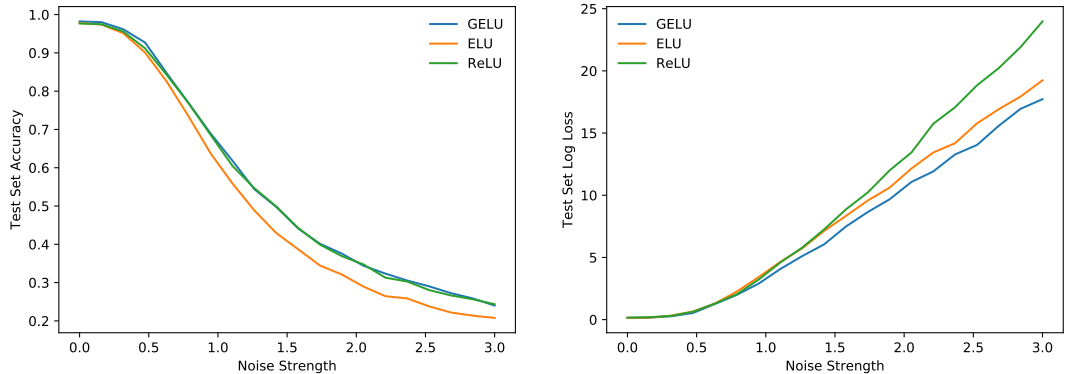

Figure 3: MNIST Robustness Results. Using different nonlinearities, we record the test set accuracy decline and log loss increase as inputs are noised. The MNIST classifier trained without dropout received inputs with uniform noise Unif$[-a, a]$ added to each example at different levels $a$, where $a = 3$ is the greatest noise strength. Here GELUs display robustness matching or exceeding ELUs and ReLUs.

network is trained for 50 epochs with a batch size of 128. This experiment differs from those of Clevert et al. in that we use the Adam optimizer (Kingma & Ba, 2015) rather than stochastic gradient descent without momentum, and we also show how well nonlinearities cope with dropout. Weights are initialized with unit norm rows, as this has positive impact on each nonlinearity's performance (Hendrycks & Gimpel, 2016; Mishkin & Matas, 2016; Saxe et al., 2014). Note that we tune over the learning rates $\{10^{-3}, 10^{-4}, 10^{-5}\}$ with 5k validation examples from the training set and take the median results for five runs. Using these classifiers, we demonstrate in Figure 3 that classifiers using a GELU can be more robust to noised inputs. Figure 2 shows that the GELU tends to have the lowest median training log loss with and without dropout. Consequently, although the GELU is inspired by a different stochastic process, it comports well with dropout.

## 3.2 MNIST AUTOENCODER

We now consider a self-supervised setting and train a deep autoencoder on MNIST (Desjardins et al., 2015). To accomplish this, we use a network with layers of width 1000, 500, 250, 30, 250, 500, 1000, in that order. We again use the Adam optimizer and a batch size of 64. Our loss is the mean squared loss. We vary the learning rate from $10^{-3}$ to $10^{-5}$. We also tried a learning rate of 0.01 but ELUs diverged, and GELUs and RELUs converged poorly. The results in Figure 4 indicate the GELU accommodates different learning rates and that the GELU either ties or significantly outperforms

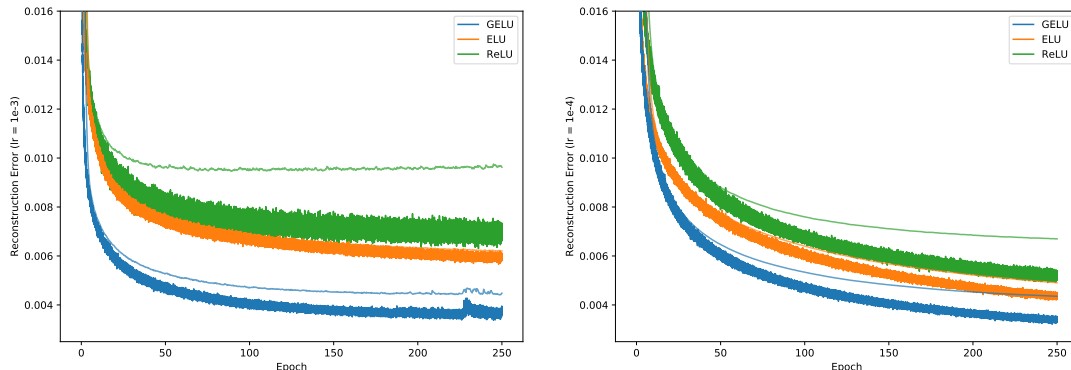

Figure 4: MNIST Autoencoding Results. Each curve is the median of three runs. Left are loss curves for a learning rate of $10^{-3}$, and the right figure is for a $10^{-4}$ learning rate. Light, thin curves correspond to test set log losses.

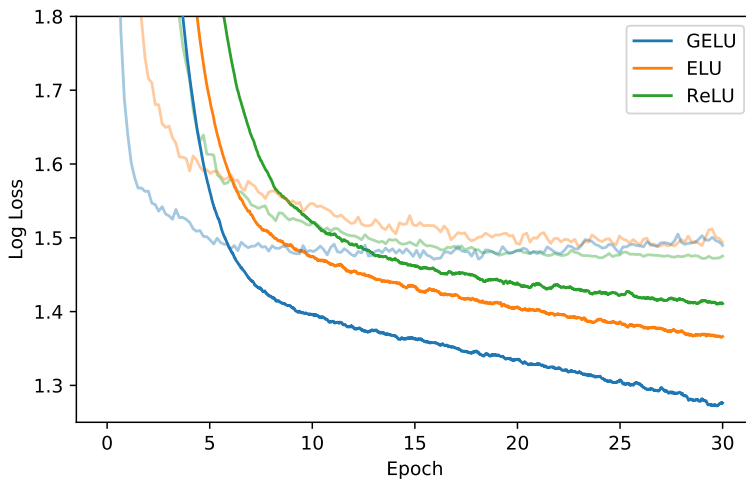

Figure 5: TIMIT Frame Classification. Learning curves show training set convergence, and the lighter curves show the validation set convergence.

the other nonlinearities. To save space, we show the learning curve for the $10^{-5}$ learning rate in appendix A.

## 3.3 TWITTER POS TAGGING

Many datasets in natural language processing are relatively small, so it is important that an activation generalize well from few examples. To meet this challenge we compare the nonlinearities on POS-annotated tweets (Gimpel et al., 2011; Owoputi et al., 2013) which contain 25 tags. The tweet tagger is simply a two-layer network with pretrained word vectors trained on a corpus of 56 million tweets (Owoputi et al., 2013). The input is the concatenation of the vector of the word to be tagged and those of its left and right neighboring words. Each layer has 256 neurons, a dropout keep probability of 0.8, and the network is optimized with Adam while tuning over the learning rates $\{10^{-3}, 10^{-4}, 10^{-5}\}$. We train each network five times per learning rate, and the median test set error is 12.57% for the GELU, 12.67% for the ReLU, and 12.91% for the ELU.

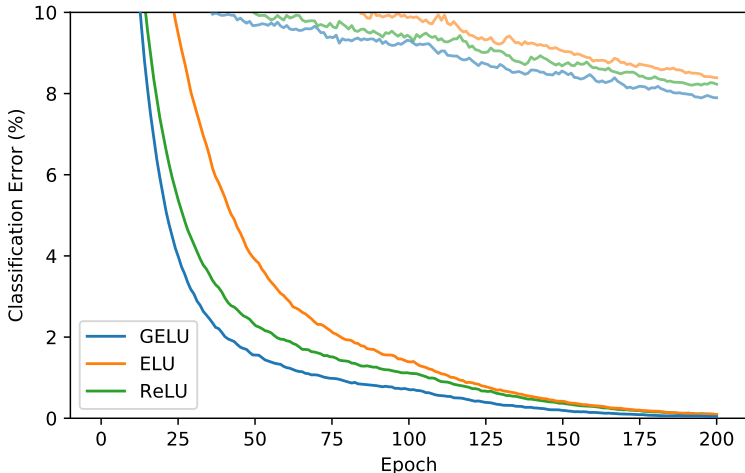

Figure 6: CIFAR-10 Results. Each curve is the median of three runs. Learning curves show training set error rates, and the lighter curves show the test set error rates.

### 3.4 TIMIT FRAME CLASSIFICATION

Our next challenge is phone recognition with the TIMIT dataset which has recordings of 680 speakers in a noiseless environment. The system is a five-layer, 2048-neuron wide classifier as in (Mohamed et al., 2012) with 39 output phone labels and a dropout rate of 0.5 as in (Srivastava, 2013). This network takes as input 11 frames and must predict the phone of the center frame using 26 MFCC, energy, and derivative features per frame. We tune over the learning rates $\{10^{-3}, 10^{-4}, 10^{-5}\}$ and optimize with Adam. After five runs per setting, we obtain the median curves in Figure 5, and median test error chosen at the lowest validation error is 29.3% for the GELU, 29.5% for the ReLU, and 29.6% for the ELU.

### 3.5 CIFAR-10/100 CLASSIFICATION

Next, we demonstrate that for more intricate architectures the GELU nonlinearity again outperforms other nonlinearities. We evaluate this activation function using CIFAR-10 and CIFAR-100 datasets (Krizhevsky, 2009) on shallow and deep convolutional neural networks, respectively.

Our shallower convolutional neural network is a 9-layer network with the architecture and training procedure from Salimans & Kingma (2016) while using batch normalization to speed up training. The architecture is described in appendix B and recently obtained state of the art on CIFAR-10 without data augmentation. No data augmentation was used to train this network. We tune over the learning initial rates $\{10^{-3}, 10^{-4}, 10^{-5}\}$ with 5k validation examples then train on the whole training set again based upon the learning rate from cross validation. The network is optimized with Adam for 200 epochs, and at the 100th epoch the learning rate linearly decays to zero. Results are shown in Figure 6, and each curve is a median of three runs. Ultimately, the GELU obtains a median error rate of **7.89**%, the ReLU obtains 8.16%, and the ELU obtains 8.41%.

Next we consider a wide residual network on CIFAR-100 with 40 layers and a widening factor of $4$ (Zagoruyko & Komodakis, 2016). We train for 50 epochs with the learning rate schedule described in (Loshchilov & Hutter, 2016) ($T_0 = 50, \eta = 0.1$) with Nesterov momentum, and with a dropout keep probability of 0.7. Some have noted that ELUs have an exploding gradient with residual networks (Shah et al., 2016), and this is alleviated with batch normalization at the end of a residual block. Consequently, we use a Conv-Activation-Conv-Activation-BatchNorm block architecture to be charitable to ELUs. Over three runs we obtain the median convergence curves in Figure 7. Meanwhile, the GELU achieves a median error of **20.74**%, the ReLU obtains 21.77% (without our changes described above, the original 40-4 WideResNet with a ReLU obtains 22.89% (Zagoruyko & Komodakis, 2016)), and the ELU obtains 22.98%.

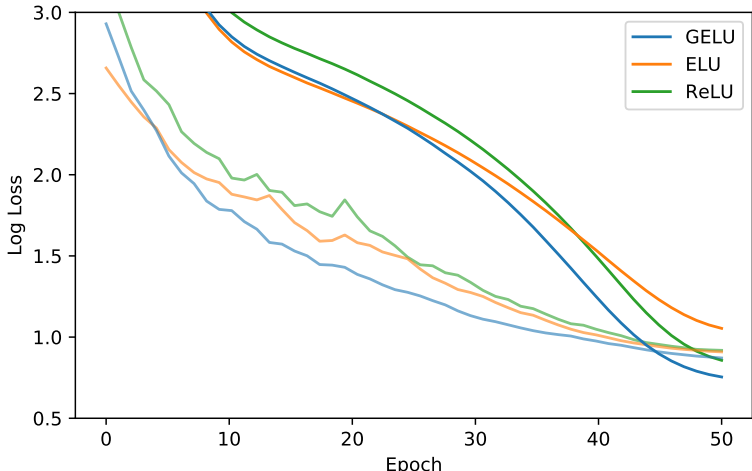

Figure 7: CIFAR-100 Wide Residual Network Results. Learning curves show training set convergence with dropout on, and the lighter curves show the test set convergence with dropout off.

## 4 SOI MAP EXPERIMENTS

Now let us consider how well the SOI Map performs rather than the GELU, its expectation. We consider evaluating the SOI Map, or an Adaptive Dropout variant without any nonlinearity, to show that neural networks do not require traditional nonlinearities. We can expect the SOI map to perform differently from a nonlinearity plus dropout. For one, stochastic regularizers applied to composed linear maps without a deterministic nonlinearity tend to yield a regularized deep linear transformation. In the case of a single linear transformation dropout and the SOI map behave differently. To see this, recall that Wang & Manning (2013) showed that for least squares regression, if a prediction is $\hat{Y} = \sum_i w_i x_i m_i$, where $x$ is an zero-centered input, $w$ is a zero-centered learned weight, and $m$ is a dropout mask of zeros and ones, we have that $\text{Var}(\hat{Y}) = \sum_i w_i^2 x_i^2 p(1-p)$ when using dropout. Meanwhile, the SOI map has the prediction variance $\sum_i w_i^2 x_i^2 \Phi(x)(1-\Phi(x))$. Thus as $x_i$ increases, the variance of the prediction increases for dropout, but for the SOI map $x_i$'s increase is dampened by the $\Phi(x)(1-\Phi(x))$ term. Then as the inputs and score gets larger, a prediction with the SOI map can have less volatility rather than more. In the experiments that follow, we confirm that the SOI map and dropout differ because the SOI map yields accuracies comparable to nonlinearities plus dropout, despite the absence of any traditional nonlinearity.

We begin our experimentation by reconsidering the 8-layer MNIST classifier. We have the same training procedure except that we tune the dropout keep probability over $\{1, 0.75, 0.5\}$ when using a nonlinearity. There is no dropout while using the SOI map. Meanwhile, for the SOI map we tune no additional hyperparameter. When the SOI map trains we simply mask the neurons, but during testing we use the expected transformation of the SOI map (the GELU) to make the prediction deterministic, mirroring how dropout is turned off during testing. A ReLU with dropout obtains 2.10% error, and a SOI map achieves 2.00% error.

Next, we reconsider the Twitter POS tagger. We again perform the same experimentation but also tune over the dropout keep probabilities $\{1, 0.75, 0.5\}$ when using a nonlinearity. In this experiment, the ReLU with dropout obtains 11.9% error, and the SOI map obtains 12.5% error. It is worth mentioning that the best dropout setting for the ReLU was when the dropout keep probability was 1, i.e., when dropout was off, so the regularization provided by the SOI map was superfluous.

Finally, we turn to an earlier TIMIT experiment. Like the previous two experiments, we also tune over the dropout keep probabilities $\{1, 0.75, 0.5\}$ when using a nonlinearity. Under this setup, the ReLU ties with the SOI map as both obtain 29.46% error, though the SOI map obtained its best validation loss in the 7th epoch while the ReLU with dropout did in the 27th epoch.

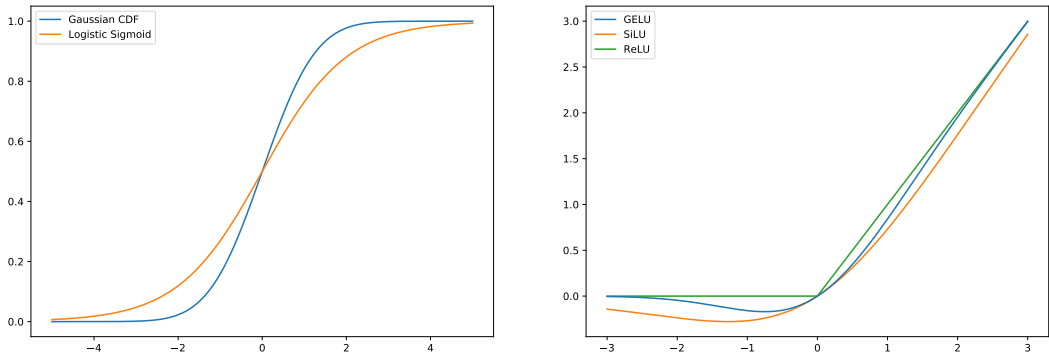

Figure 8: Although a logistic sigmoid function approximates a Gaussian CDF, the difference is still conspicuous and is not a suitable approximation.

In summary, the SOI map can be comparable to a nonlinearity with dropout and does not simply yield a regularized linear transformation. This is surprising because the SOI map is not like a traditional nonlinearity while it has a nonlinearity's power. The upshot may be that traditional, deterministic, differentiable functions applied to a neuron's input are less essential to the success of neural networks, since a stochastic regularizer can achieve comparable performance.

## 5 DISCUSSION

Across several experiments, the GELU outperformed previous nonlinearities, but it bears semblance to the ReLU and ELU in other respects. For example, as $\sigma \to 0$ and if $\mu = 0$, the GELU becomes a ReLU. More, the ReLU and GELU are equal asymptotically. In fact, the GELU can be viewed as a natural way to smooth a ReLU. To see this, recall that ReLU $= \max(x,0) = x\mathbb{1}(x > 0)$ (where $\mathbb{1}$ is the indicator function), while the GELU is $x\Phi(x)$ if $\mu = 0, \sigma = 1$. Then the CDF is a smooth approximation to the binary function the ReLU uses, like how the sigmoid smoothed binary threshold activations. Unlike the ReLU, the GELU and ELU can be both negative and positive. In fact, if we used the cumulative distribution function of the standard Cauchy distribution, then the ELU (when $\alpha = 1/\pi$) is asymptotically equal to $xP(C \le x), C \sim \mathsf{Cauchy}(0,1)$ for negative values and for positive values is $xP(C \le x)$ if we shift the line down by $1/\pi$. These are some fundamental relations to previous nonlinearities.

However, the GELU has several notable differences. This non-convex, non-monotonic function is not linear in the positive domain and exhibits curvature at all points. Meanwhile ReLUs and ELUs, which are convex and monotonic activations, are linear in the positive domain and thereby can lack curvature. As such, increased curvature and non-monotonicity may allow GELUs to more easily approximate complicated functions than can ReLUs or ELUs. Also, since ReLU$(x) = x\mathbb{1}(x > 0)$ and GELU$(x) = x\Phi(x)$ if $\mu = 0, \sigma = 1$, we can see that the ReLU gates the input depending upon its sign, while the GELU weights its input depending upon how much greater it is than other inputs. In addition and significantly, the GELU has a probabilistic interpretation given that it is the expected SOI map, which combines ideas from dropout and zoneout.

The SOI Map also relates to a previous stochastic regularizer called Adaptive Dropout (Ba & Frey, 2013). The crucial difference between typical adaptive dropout and the SOI map is that adaptive dropout multiplies the nonlinearity's output by a mask, but the SOI map multiplies the neuron input by a mask. Consequently, the SOI map trains without any nonlinearity, while adaptive dropout modifies the output of a nonlinearity. In this way, standard implementations of adaptive dropout do not call into question the necessity of traditional nonlinearities since it augments a nonlinearity's decision rather than eschews the nonlinearity entirely.

We also have two practical tips for using the GELU. First we advise using an optimizer with momentum when training with a GELU, as is standard for deep neural networks. Second, using a close approximation to the cumulative distribution function of a Gaussian distribution is important. For

example, using a sigmoid function $\sigma(x) = 1/(1 + e^{-x})$ is an approximation of a cumulative distribution function of a normal distribution, but it is not a close enough approximation (Ba & Frey, 2013). Indeed, we found that a Sigmoid Linear Unit (SiLU) $x\sigma(x)$ performs worse than GELUs but usually better than ReLUs and ELUs. The maximum difference between $\sigma(x)$ and $\Phi(x)$ is approximately 0.1, but the difference between the two is visible in Figure 8. Instead of using a $x\sigma(x)$ to approximate $\Phi(x)$, we used $0.5x(1 + \tanh[\sqrt{2/\pi}(x + 0.044715x^3)])$ (Choudhury, 2014).[1] This is a sufficiently fast, easy-to-implement approximation which we used in every experiment in this paper.

## 6 CONCLUSION

We observed that the GELU outperforms previous nonlinearities across tasks from computer vision, natural language processing, and automatic speech recognition. Moreover, we showed that a stochastic regularizer can compete with a nonlinearity aided by dropout, indicating that traditional nonlinearities may not be crucial to neural network architectures. This stochastic regularizer makes probabilistic decisions and the GELU is the expectation of the decision. We therefore probabilistically related the GELU to the SOI map, thereby bridging a nonlinearity to a stochastic regularizer. Now having seen that a stochastic regularizer can replace a traditional nonlinearity, we hope that future work explores the design space of other stochastic regularizers as powerful as a traditional activation aided by dropout. Furthermore, there may be fruitful modifications to the GELU in different contexts. For example, for sparser inputs, a nonlinearity of the form $xP(L \leq x), L \sim \mathsf{Laplace}(0, 1)$ may be a more effective activation. For the numerous datasets evaluated in this paper, the GELU exceeded the accuracy of the ELU and ReLU consistently, making it a viable alternative to previous nonlinearities.

## ACKNOWLEDGMENT

We would like to thank NVIDIA Corporation for donating several TITAN X GPUs used in this research.

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

## A    ADDITIONAL MNIST AUTOENCODER LEARNING CURVE

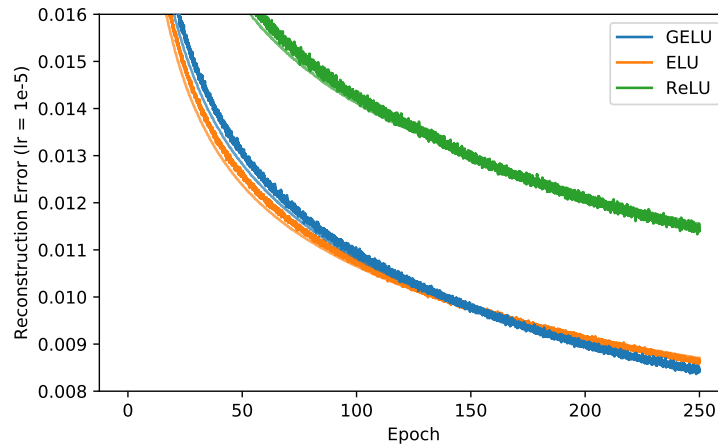

Figure 9: MNIST Autoencoding Results for a learning rate of $10^{-5}$. Each curve is a median of three runs. Light, thin curves correspond to test set log losses. Note that reconstruction errors are higher than models trained with $10^{-3}$ or $10^{-4}$ learning rates.

## B    NEURAL NETWORK ARCHITECTURE FOR CIFAR-10 EXPERIMENTS

Table 1: Neural network architecture for CIFAR-10.

| Layer Type | # channels | $x, y$ dimension |
|---|---|---|
| raw RGB input | 3 | 32 |
| ZCA whitening | 3 | 32 |
| Gaussian noise $\sigma = 0.15$ | 3 | 32 |
| $3 \times 3$ conv with activation | 96 | 32 |
| $3 \times 3$ conv with activation | 96 | 32 |
| $3 \times 3$ conv with activation | 96 | 32 |
| $2 \times 2$ max pool, stride 2 | 96 | 16 |
| dropout with $p = 0.5$ | 96 | 16 |
| $3 \times 3$ conv with activation | 192 | 16 |
| $3 \times 3$ conv with activation | 192 | 16 |
| $3 \times 3$ conv with activation | 192 | 16 |
| $2 \times 2$ max pool, stride 2 | 192 | 8 |
| dropout with $p = 0.5$ | 192 | 8 |
| $3 \times 3$ conv with activation | 192 | 6 |
| $1 \times 1$ conv with activation | 192 | 6 |
| $1 \times 1$ conv with activation | 192 | 6 |
| global average pool | 192 | 1 |
| softmax output | 10 | 1 |

