# Peer review of "Bridging Nonlinearities and Stochastic Regularizers with Gaussian Error Linear Units"

_ICLR 2017 — rejected_

[Author Response · Dan Hendrycks · 15 Dec 2016]
**A Brief Comment on the SOI Map**

It is worth mentioning that the SOI map is not a proposed dropout replacement--we only conclude that it is "comparable to nonlinearities plus dropout," as our experiments show. In the current draft, we even call it "an Adaptive Dropout variant without any nonlinearity." We mention the SOI map only because it aids in our motivation of the GELU, shows that traditional nonlinearities are not necessary for training, and because it is an endpoint of a bridge from a stochastic regularizer to a nonlinearity. We do not intend for it to be construed as a proposed dropout replacement, and I am sorry if it was.

[Official Review · AnonReviewer2 · rating 5 · confidence 4 · 15 Dec 2016]
**The proposed approach seems similar to other existing approaches in literature (eg. adaptive dropout). Experimental validation not adequate for evaluation.**

Approaches like adaptive dropout also have the binary mask as a function of input to a neuron very similar to the proposed approach. It is not clear, even from the new draft, how the proposed approach differs to Adaptive dropout in terms of functionality. The experimental validation is also not extensive since comparison to SOTA is not included.

[Official Review · AnonReviewer3 · rating 4 · confidence 4 · 16 Dec 2016]
**Minor variant on existing regularization methods**

The proposed regularizer seems to be a particular combination of existing methods. Though the implied connection between nonlinearities and stochastic regularizers is intriguing, in my opinion the empirical performance does not exceed the performance achieved by similar methods by a large enough margin to arrive at a meaningful conclusion.

[Official Review · AnonReviewer1 · rating 5 · confidence 4 · 16 Dec 2016]
**Official review.**

The method proposed essential trains neural networks without a traditional nonlinearity, using multiplicative gating by the CDF of a Gaussian evaluated at the preactivation; this is motivated as a relaxation of a probit-Bernoulli stochastic gate. Experiments are performed with both.

The work is somewhat novel and interesting. Little is said about why this is preferable to other similar parameterizations of the same (sigmoidal? softsign? etc.) It would be stronger with more empirical interrogation of why this works and exploration of the nearby conceptual space. The CIFAR results look okay by today's standards but the MNIST results are quite bad, neural nets were doing better than 1.5% a decade ago and the SOI map results (and the ReLU baseline) are above 2%. (TIMIT results on frame classification also aren't that interesting without evaluating word error rate within a speech pipeline, but this is a minor point.)

The idea put forth that SOI map networks without additional nonlinearities are comparable to linear functions is rather misleading as they are, in expectation, nonlinear functions of their input. Varying an input example by multiplying or adding a constant will not be linearly reflected in the expected output of the network. In this sense they are more nonlinear than ReLU networks which are at least locally linear.

The plots are very difficult to read in grayscale,

[Final Decision · Program Chairs · 06 Feb 2017]
**ICLR committee final decision**

The reviewers unanimously recommend rejecting the paper.